# Associations among Health Status, Occupation, and Occupational Injuries or Diseases: A Multi-Level Analysis

**DOI:** 10.3390/diagnostics13030381

**Published:** 2023-01-19

**Authors:** Shu-Yuan Su, Yu-Wen Li, Fur-Hsing Wen, Chi-Yu Yao, Jong-Yi Wang

**Affiliations:** 1Public Health, China Medical University, Taichung 406040, Taiwan; 2Department of Human Resource, Wu Feng Cheng Ching Hospital, Taichung 412031, Taiwan; 3School of Business, Soochow University, Taipei 100006, Taiwan; 4Department of Psychiatry, An Nan Hospital, Tainan City 709204, Taiwan; 5Department of Health Services Administration, China Medical University, Taichung 406040, Taiwan

**Keywords:** occupational injury and disease, occupation, binary logistic regression (BLR), hierarchical generalized linear model (HGLM)

## Abstract

Purpose: The present study used a hierarchical generalized linear model to explore the effects of physical and mental health and occupational categories on occupational injuries and diseases. Methods: The data were obtained from the Registry for Beneficiaries of the 2002–2013 National Health Insurance Research Database. The benefit categories involved adults with occupational injuries and diseases. Six major occupational categories and 28 subcategories were used. The main analysis methods were binary logistic regression (BLR) and hierarchical generalized linear model (HGLM). Results: After adjustment for relevant factors, the three major occupation subcategories most likely to develop occupational injuries and diseases were Subcategory 12 “employees with fixed employers” of Category 1 “civil servants, employees in public or private schools, laborers, and self-employed workers”; Subcategory 2 “employees in private organizations” of Category 1; and “sangha and religionists” of Category 6 “other citizens.” Conditions such as mental disorders and obesity increased the risk of occupational injuries and diseases. Conclusion: A portion of the occupational categories had a higher risk of occupational injuries and diseases. Physical and mental health issues were significantly correlated with occupational injuries and diseases. To the authors’ knowledge, this is the first study to use HGLM to analyze differences in occupational categories in Taiwan.

## 1. Introduction

The Occupational Safety and Health Administration, Ministry of Labor (2014) of Taiwan differentiates between occupational injuries and occupational diseases as follows: Occupational injuries refer to immediate accidental injuries while executing duties, whereas occupational diseases refer to physician-diagnosed diseases resulting from exposure to chemical, physical, biological, ergonomic, or other factors while executing duties. Accidents, including occupational injuries, are the seventh leading cause of death in Taiwan [1].

Studies on occupational injuries and diseases have focused on workplace problems related to work environments or laws [2,3], but few have explored the effects of personal health factors on occupational injuries and diseases. However, recent international studies have reported that obesity may increase the risk of occupational injuries [4,5], as does depression, chronic diseases, and the use of medications for specific diseases, such as hypertension [6]. Studies have explored the relations between various occupation categories and occupational injuries and diseases [7,8], and those from Taiwan have mainly used the six major categories of the insured in the National Health Insurance Research Database as occupation categories [9,10]. However, no study has used the 28 occupational subcategories provided by the National Health Insurance Administration for consideration and research.

To mitigate the research gap, this study involved rigorous and complete analysis of factors affecting occupational injuries and diseases and discussion of the effects of personal physical and mental health factors and occupational categories on occupational injuries and diseases. The occupational categories for discussion consisted of the 28 occupational subcategories provided by the National Health Insurance Administration (e.g., civil servants, employees in schools, and soldiers). To facilitate data analysis by considering different levels of influential variables, this study included the various levels of influential variables in a generalized hierarchical linear model (HGLM) and considered the influence of such variables on occupational injuries and diseases to evaluate how the differences among groups, organizations, and contexts affect individual results [11,12]. Subsequently, the quality of laborers can be improved and safety professional talent can be cultivated to effectively reduce the incidence of occupational injury accidents [13,14,15]. Taiwan National Health Insurance Research Database Insurance costs are proportional to wages. This insurance is compulsory social insurance. The present study is the first in Taiwan to use HGLM to analyze national representative data and explore how physical and mental health status and occupational categories are correlated with occupational injuries and diseases will help reduce occupational accidents by understanding the impact of diseases on labor.

## 2. Materials and Methods

In 2016, this study archived 1 million samples using the Taiwan’s National Health Insurance Research Database released by the National Health Research Institutes (NHRI) in 2005. Occupational injury and illness benefit codes from the 2002–2013 database were applied to extract employed adults aged ≥20 years. A total of 529,278 patients were included in the sample. Among them, “underwriting information sheet (ID)”, “outpatient prescription and treatment list (CD)”, “inpatient medical expense list (DD)”, and “outpatient prescription doctor’s order list (OO)” were extracted as analysis data sources. The current study was approved by the Research Ethics Committee of China Medical University and Hospital, Taichung, Taiwan (CMUH REC No: N/A/CRREC-104-012).

Binary logistic regression (BLR) and HGLM were conducted using SAS 9.3 software. The BLR was used to analyze the correlation between the sample characteristics and the development of occupational injuries and diseases. The odds ratio was used to compare the risk of occupational injuries and diseases between groups. HGLM was used to explore the interdependence between micro-level (individual) factors (demographic characteristics and health status) under macro-level factors (occupational categories and subcategories). This process avoided individualistic fallacy in deduction resulting from a nested structure of data, while indicating whether significant differences exist between macro- and micro-level factors [16]. In summary, HGLM was more suitable for analyzing the data of the present study than BLR. In addition, variance inflation factors (VIFs) were used to determine whether collinearity existed between influencing variables [17].

The study framework is illustrated in Figure 1. The macro-level unit comprised six occupational categories: Category 1 included civil servants, employees in public or private schools, laborers, and self-employed workers; Category 2 included professional workers, seamen, and captains; Category 3 included farmers and fishermen; Category 4 included soldiers; Category 5 included individuals from low-income households; and Category 6 included sangha and other citizens. The number of occupational subcategories was 28. The ages of individual level units were divided into five groups, and the insured amount was divided into six levels. Physical and mental health status was assessed according to whether the individual was with mental disorders, diabetes, cardiovascular diseases, obesity, hypertension, or asthma, and whether the individual used sedative–hypnotic drugs, antipsychotics, controlled analgesics, cardiovascular drugs, and diuretics. (Table 1 and Table 2).

## 3. Results

This study used VIFs to determine whether collinearity existed between the variables, with a VIF value >10 indicating potential collinearity. The results revealed that the highest VIF value was 1.366, implying no collinearity between the influential variables.

BLRs for each influential variable in relation to the development of occupational injuries and diseases were established to clearly analyze the effect and risk of influential variables on the development of occupational injuries and diseases. The results of the BLR revealed the following.

In the macro-level units, when Category 1 was used as the reference group, the other five categories presented significance in whether suffering from occupational injury and diseases. Compared with Category 1, the risk of occupational injuries and diseases in Category 2 was 0.936-fold lower and in Category 3 was 3.808-fold higher. When the subcategory of civil servants of the central governmental agencies was used as the reference group, 14 of the remaining 28 subcategories exhibited significance. Compared with the reference group, the risk of occupational injuries and diseases was lower in military school students and compulsory servicemen (0.161-fold), employees in private business organizations (0.267-fold), employees of nonprofit organizations (0.273-fold), and members of occupational unions (0.286-fold). By contrast, employees with a fixed employer and veterans who were insured through township offices had 3.051-fold and 3.217-fold higher risks, respectively, than the reference group. Civil servants, labor, and self-employed (*p* < 0.001, OR = 1.402). Due to the large labor force of grassroots workers at schools and provincial (city) agencies and agencies below the level, the employment risk is high.

In the micro-level units regarding demographic characteristics, compared with women, men had a significantly lower risk (0.703-fold) of occupational injuries and diseases. When the age group between 20 and 30 years was used as reference, the remaining four age groups exhibited significance. At the same time, compared with the reference group, the age group of 31–40 years had a 1.154-fold higher risk of occupational injuries and diseases, whereas the age group of ≥60 years had a 14.337-fold higher risk. When the group with an insured amount of NTD 22,800 or lower was used as reference, the other five range groups exhibited significance. The group with an insured amount of NTD 36,301–45,800 had 0.177-fold lower risks, whereas the group with an insured amount of NTD 57,801 or higher had 0.264-fold higher risks. It can be seen that the amount of insurance is high, and the work pressure of the supervisor is high.

In the micro-level units regarding physical and mental health status, six conditions and five drugs were considered in this study. Individuals with mental disorders, obesity, hypertension, and asthma, as well as antipsychotic or cardiovascular drug use exhibited significance. The risk of occupational injuries and diseases among individuals with mental disorders was 0.802-fold lower than among those without mental disorders. The risk of occupational injuries and diseases among individuals with obesity was 0.887-fold lower than among those without obesity. The risk of occupational injuries and diseases among individuals with hypertension was 1.040-fold higher than among those without hypertension. The risk of occupational injuries and diseases among individuals with asthma was 0.864-fold lower than those without asthma. The risk of occupational injuries and diseases among those using antipsychotics was 0.845-fold lower than among those without antipsychotic use. The risk of occupational injuries and diseases among individuals who used cardiovascular drugs was 0.823-fold lower than among those without cardiovascular drug use.

Subsequently, HGLMs of demographic characteristics, physical and mental health status, and occupational categories in relation to occupational injuries and diseases were established to examine how the influential variables of the macro- and micro-level units affect occupational injuries and diseases of individuals. The results of the HGLM are as follows.

In the macro-level units, after relevant factors of the micro-levels, including sex, age, disease (conditions), and drugs taken, were controlled for, the occupation subcategories that were prone to develop occupational injuries and diseases were employees with fixed employers (Subcategory 12 in Category 1), civil servants in provincial (city) or lower levels of governmental agencies (Subcategory 2 in Category 1), and sangha or religionists of Category 6; these subcategories had 4.1074-fold, 2.0726-fold, and 2.0079-fold risks, respectively, compared with the reference group. The members of occupational unions (Subcategory 17 in Category 2), employees of nonprofit organizations (Subcategory 13 in Category 1), employees in provincial (city) or lower levels of governmental agencies or schools (Subcategory 10 in Category 1), and employees in small private business organizations (Subcategory 8 in Category 1) had lower risks compared with the reference group (0.3809-fold, 0.4296-fold, 0.4354-fold, and 0.4475-fold, respectively).

In the micro-level units regarding demographic characteristics, when female was used as reference, the male group exhibited a significantly lower risk of occupational injuries and diseases (0.631-fold compared with the reference group). When the age group of 20–30 years was used as reference, the other four age groups exhibited significance. At the same time, among these four significant age groups, the age group of 31–40 years (1.172-fold compared with the reference group) had relative lower risks of developing occupational injuries and diseases, whereas the age group of ≥60 years (14.563-fold compared with the reference group) had relative higher risks. When the group with an insured amount of NTD 22,800 or lower was used as reference, four out of the other five range groups exhibited significance. Compared with the reference group, the group with an insured amount of NTD 36,301–45,800 had 0.914-fold lower risks, whereas the group with an insured amount of NTD 57,801 or higher had 1.386-fold higher risks.

In the micro-level units regarding physical and mental health status, when the group without mental disorders was used as reference, the risk of occupational injuries and diseases among individuals with mental disorders was 1.258-fold higher. When the group without diabetes was used as reference, the risk of occupational injuries and diseases among individuals with diabetes was 1.997-fold higher. When the group without chronic heart disease was used as reference, the risk of occupational injuries and diseases among individuals with chronic heart disease was 1.004-fold higher. When the group without obesity was used as reference, the risk of occupational injuries and diseases among individuals with obesity was 1.138-fold higher. When the group without hypertension was used as reference, the risk of occupational injuries and diseases among individuals with hypertension was 1.965-fold higher. When the group without asthma was used as reference, the risk of occupational injuries and diseases among individuals with asthma was 1.138-fold higher. When the group of individuals who did not use sedative–hypnotic drugs was used as reference, the risk of occupational injuries and diseases among individuals who used sedative–hypnotic drugs was 1.076-fold higher. When the group of individuals who did not take antipsychotics was used as reference, the risk of occupational injuries and diseases among those who used antipsychotics was 1.844-fold higher. When the group of individuals who did not use controlled analgesics was used as reference, the risk of occupational injuries and diseases among individuals who used controlled analgesics was 1.060-fold higher. When the group of individuals who did not use cardiovascular drugs was used as reference, the risk of occupational injuries and diseases among individuals who used cardiovascular drugs was 1.818-fold higher. When the group of individuals who did not use diuretics was used as reference, the risk of occupational injuries and diseases among individuals who used diuretics was 1.889-fold higher.

## 4. Discussion

Compared with studies that focused on workplace factors, the present study used demographic characteristics, health status, and occupational categories as the factors to explore their relevant risks and effects on occupational injuries and diseases. All of the medical and mental disorders could not be included in this study. The results summarizing previous analyses based mainly on HGLM revealed the following.

Comparison results between BLR and HGLM are presented in Table 3. In terms of demographic characteristics and physical and mental health status, HGLM yielded more diverse and accurate results regarding the significant effects on occupational injuries and diseases compared with BLR, thereby helping to understand the distribution of the two levels of influential variables concerning occupational injuries and diseases.

The results indicated that sex, age, and the insured amount had significant effects on occupational injuries and diseases. In addition, women are more prone to work-related injuries and occupational diseases than men due to their weaker physique. This study is consistent with literature reports [13,14,18]. In this study, the risk of occupational inquiries and diseases increased with age; by contrast, a study claimed that people aged 31–40 years had the highest risk [15]. Moreover, compared with individuals with high income levels, those with low income levels had a higher risk of occupational injuries and diseases, possibly because they had lower socioeconomic statuses and had occupations with higher risks of occupational hazards [19,20].

The results regarding physical and mental health status indicated that influential variables, including diabetes, obesity, asthma, chronic heart diseases, and mental disorders, exhibited significant correlations with the development of occupational injuries and diseases, in agreement with the results of domestic and international studies [4,6,17]. Obesity is okay, but obesity increases the risk of comorbidities, disease, and injury. In addition, the use of sedative–hypnotic drugs, antipsychotics, controlled analgesics, cardiovascular drugs, and diuretics exerted significant effects on the development of occupational injuries and diseases, in agreement with the results of previous studies [21,22,23,24].

The results regarding occupational categories indicated that employees (laborers) of public business organizations (Subcategory 7 in Category 1) had a significant effect on the development of occupational injuries and diseases. However, the employees (civil servants) of public business organizations (Subcategory 6 in Category 1) did not have a significant effect. The results revealed that common laborers, because of their work contents, were more likely to develop occupational injuries and diseases compared with civil servants. The civil servants in provincial (city) or lower levels of governmental agencies (Subcategory 2 in Category 1) and employees in private second or primary schools (Subcategory 5) had significant effects on the development of occupational injuries and diseases. Their assistance to students’ use of equipment and daily labor-oriented physical injuries were estimated to be the main causes. In Category 6, religionists and sangha had significant effects on the development of occupational injuries and diseases. A study indicated that older populations of the same ethnicity and cultures in Thailand exhibited a significantly higher prevalence of knee joint pain and arthritis among Buddhists than among people of other religions [25,26,27].

## 5. Conclusions

According to the analysis results and discussion, the present study proposes the following concrete suggestions for governmental agencies and medical institutions:For the occupational subcategories of employees with fixed employers (Subcategory 12 in Category 1), employees of private business organizations (Subcategory 2), and sangha and religionists of Category 6, who were prone to occupational injuries and diseases, the government shall prioritize stipulation of relevant health policy and regulations to facilitate adequate monitoring and intervention.Older age as well as long-term exposure to chemical substances increase the risk of occupational injuries and diseases. Therefore, relevant agencies must establish norms to allow free and periodic health examinations for laborers with a certain period of seniority to prevent occupational injuries and diseases.Employers must ensure occupational environment safety and health. They should conduct on-the-job training and establish rapid occupational injury report systems to minimize the incidence and reduce the severity of occupational accidents.Occupational injury clinics should implement integrated medical procedures. In addition, occupational injury prevention centers shall be established to reduce the time to access medical care in case of occupational injuries.Health institutes should periodically monitor patients of occupational accidents or patients with chronic diseases (e.g., hypertension and cardiovascular diseases), remind these patients of the medication safety and side effects that may affect work, and arrange periodic health examinations for the patients to control their conditions.

This study used HGLM to analyze influential variables from different level units and attained three contributions. First, the results helped understand the influence of Taiwanese citizens’ demographic characteristics, physical and mental health status, and occupational categories on the risk of occupational injuries and diseases. Second, HGLM was used to analyze occupational categories and subcategories to mitigate the literature gap. Whether for the use of HGLM or occupational subcategories, both are the first for the field of occupational injuries and disease research in Taiwan. Finally, the study findings can guide health departments and labor agencies in Taiwan in establishing labor protection and policy planning regulations and preventing occupational injuries and diseases [28,29,30].

## Figures and Tables

**Figure 1 diagnostics-13-00381-f001:**
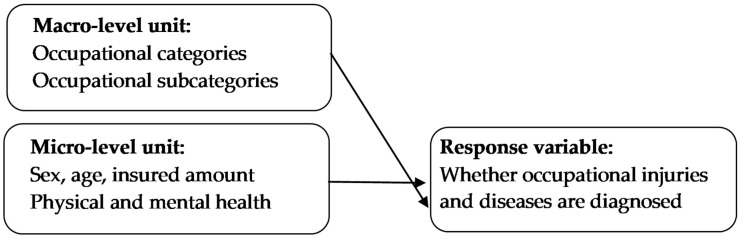
Research framework for exploring the effects of physical and mental health status and occupational categories on occupational injuries and diseases.

**Table 1 diagnostics-13-00381-t001:** HGML results for health status and occupational injuries or diseases.

Variable	Presence of Occupational Injuries or Diseases		
	OR	95% C.I.	*p*-Value
Sex			
Women (reference group)	--	--	--
Men	0.631	(0.598–0.664)	<0.0001 *
Age			
20–30 years (reference group)	--	--	--
31–40 years	1.172	(1.084–1.267)	<0.0001 *
41–50 years	1.247	(1.068–1.456)	<0.0001 *
51–60 years	2.913	(2.386–3.558)	0.0053 *
>60 years	14.563	(11.503–18.437)	<0.0001 *
Payroll bracket			
NTD 22,800 or lower (reference group)	--	--	--
NTD 22,801–28,800	1.353	(1.067–1.716)	0.0125 *
NTD 28,801–36,300	0.958	(0.926–0.991)	0.0129 *
NTD 36,301–45,800	0.914	(0.874–0.956)	<0.0001 *
NTD 45,801–57,800	0.997	(0.931–1.067)	0.9241
NTD 57,801 or higher	1.386	(1.315–1.461)	<0.0001 *
Urbanization level			
Highly urbanized cities or counties (reference group)	--	--	--
Moderately urbanized cities or counties	1.062	(0.997–1.131)	0.0625
Townships or county-administered cities	0.881	(0.812–0.956)	0.0025 *
Aging cities or counties	1.504	(0.956–1.162)	0.2930
Remote townships	1.060	(0.885–1.269)	0.5278
Emerging cities or counties	1.730	(1.479–2.025)	<0.0001 *
Agricultural cities or counties	1.169	(1.033–1.323)	0.0132 *
NHI Administration division			
Taipei Division (reference group)	--	--	--
Central Division	1.325	(1.277–1.375)	<0.0001 *
Northern Division	0.836	(0.781–0.895)	<0.0001 *
Eastern Division	1.152	(1.078–1.231)	<0.0001 *
Southern Division	1.069	(1.024–1.117)	0.0025*
Kaoping Division	0.585	(0.529–0.647)	<0.0001 *
Illness			
Mental illness			
Without mental illness	--	--	--
With mental illness	1.258	(1.209–1.309)	<0.0001 *
Obesity			
Not obese	--	--	--
Obese	1.138	(1.088–1.191)	<0.0001 *
Diabetes			
Without diabetes	--	--	--
With diabetes	1.997	(1.974–1.997)	<0.0001 *
Asthma			
Without asthma	--	--	--
With asthma	1.138	(1.088–1.191)	<0.0001 *
Chronic heart disease			
Without chronic heart disease	--	--	--
With chronic heart disease	1.004	(1.004–1.1.024)	<0.0001 *
Hypertension			
Without hypertension	--	--	--
With hypertension	1.965	(1.948–1.983)	0.0001 *
Medication			
Sedative–hypnotics			
Not taking sedative–hypnotics	--	--	--
Taking sedative–hypnotics	1.076	(1.106–1.140)	0.0127 *
Antipsychotics			
Not taking antipsychotics	--	--	--
Taking antipsychotics	1.844	(1.760–1.938)	0.0017 *
Controlled analgesics			
Not taking controlled analgesics	--	--	--
Taking controlled analgesics	1.060	(0.895–1.255)	0.0012 *
Cardiovascular medications			
Not taking cardiovascular medications	--	--	--
Taking cardiovascular medications	1.818	(1.794–1.842)	<0.0001 *
Diuretics			
Not taking diuretics	--	--	--
Taking diuretics	1.889	(1.791–1.999)	0.0073 *
Surgery			
Having not undergone surgery	--	--	--
Having undergone surgery	1.709	(1.609–1.815)	<0.0001 *

* *p* < 0.05.

**Table 2 diagnostics-13-00381-t002:** HGLM results for the presence of occupational injuries or diseases.

Variable	*p*-Value	OR	95% Confidence Limits
Intercept (reference groups)	--	--	--
(civil servants, labor, workers, and self-employed owners of businesses: civil servants at central agencies)	0.0133 *	1.4657	(1.0828–1.9840)
(civil servant, labor, and self-employed owners of businesses: civil servants at provincial (city) agencies and agencies below the level)	<0.0001 *	2.0726	(1.5323–2.8030)
(civil servant, labor, and self-employed owners of businesses: civil servants at local agencies)	0.7269	0.9127	(0.5466–1.5239)
(civil servant, labor, and self-employed owners of businesses: employees of private junior colleges and schools)	0.0875	1.4706	(0.9448–2.2890)
(civil servant, labor, and self-employed owners of businesses: teachers of private high and elementary schools)	0.0300 *	1.7477	(1.0557–2.8936)
(civil servant, labor, and self-employed owners of businesses: entry-level workers at publicly owned enterprises and institutions [public employee insurance program])	0.9923	1.0015	(0.7386–1.3580)
(civil servant, labor, and self-employed owners of businesses: entry-level workers at public owned enterprises and institutions [labor insurance program])	0.0002 *	0.5780	(0.4320–0.7735)
(civil servant, labor, and self-employed owners of businesses: employees of privately owned enterprises and institutions)	<0.0001 *	0.4475	(0.3405–0.5880)
civil servant, labor, and self-employed owners of businesses: entry-level workers at central agencies and national junior colleges)	<0.0001 *	0.5177	(0.3749–0.7149)
(civil servant, labor, and self-employed owners of businesses: entry-level workers at schools and provincial (city) agencies and agencies below the level)	<0.0001 *	0.4354	(0.3267–0.5805)
(civil servant, labor, and self-employed owners of businesses: entry-level workers at private schools)	0.1817	0.7034	(0.4197–1.1789)
(civil servant, labor, and self-employed owners of businesses: employees employed by particular employers)	<0.0001 *	4.1074	(2.8210–5.9799)
(civil servant, labor, and self-employed owners of businesses: employees of nonprofit enterprises and institutions)	<0.0001 *	0.4296	(0.3229–0.5716)
(civil servant, labor, and self-employed owners of businesses: independently practicing professionals and technicians)	0.9114	1.0721	(0.3147–3.6517)
(professionals, seamen, and sea captains: members of an occupational union)	<0.0001 *	0.3809	(0.2897–0.5009)
(professionals, seamen, and sea captains: seamen serving on foreign vessels who are members of the National Seamen’s Union or the Master Mariners’ Association)	0.6969	1.2571	(0.3976–3.9749)
(farmers and fishermen: farmers)	0.0121 *	1.4310	(1.0817–2.9496)
(farmers and fishermen: members of the Irrigation Association)	0.6014	1.3492	(0.4386–1.5506)
(farmers and fishermen: members of the National Fishermen’s Association)	0.0778	0.7671	(0.5713–1.7706)
(military personnel: bereaved family members of military personnel who are receiving pensions due to the death of the military personnel members and military personnel’s dependents who lost their support)	0.0389 *	0.6987	(0.4972–1.6441)
(military personnel: military school students who receive grants from the government and people who are in mandatory military service)	0.0363 *	0.5423	(0.30571.3576)
(military personnel: people who are in alternative military service)	0.9363	1.0518	(0.3053–1.3570)
(members of low-income families: members of low-income families who are placed in social welfare service institutions)	0.2500	1.5536	(0.7334–2.0821)
(members of low-income families: members of low-income families whose group insurance applicants are village [township, municipal, or district] administration offices)	0.1220	0.7772	(0.5646–1.7587)
(members of the Sangha and other Taiwanese nationals: veterans placed in social welfare service institutions)	0.2587	1.6361	(0.6964–2.0064)
(members of the Sangha and other Taiwanese nationals: veterans and bereaved family members of veterans)	0.9363	0.7893	(0.5986–1.8195)
members of the Sangha and other Taiwanese nationals: members of the Sangha and religious workers	0.0104 *	2.0079	(1.1777–3.2470)
residents of social welfare service institutions	0.5121	1.4447	(0.4810–1.6177)

* *p* < 0.05.

**Table 3 diagnostics-13-00381-t003:** Results for significant influential variables in binary logistic regression (BLR) and generalized hierarchical linear model (HGLM).

Variable	Reference Group	BLR	HGLM
Sex	Female	✓	✓
Age	20–30 years	✓	✓
Insured amount	NTD 22,800 or lower	✓	✓
Mental disorders	Without	✓	✓
Obesity	Without	✓	✓
Diabetes	Without		✓
Chronic heart diseases	Without		✓
Hypertension	Without	✓	✓
Asthma	Without	✓	✓
Sedative–hypnotic drugs	Did not use		✓
Antipsychotics	Did not use	✓	✓
Controlled analgesics	Did not use		✓
Cardiovascular drugs	Did not use	✓	✓
Diuretics	Did not use		✓

## Data Availability

The datasets used in this study are available from the Ministry of Health and Welfare, Taiwan, on reasonable request.

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
