# Peer review of "Associations among Health Status, Occupation, and Occupational Injuries or Diseases: A Multi-Level Analysis"

_diagnostics, 2023, doi:10.3390/diagnostics13030381_

Round 1
Reviewer 1 Report
The manuscript has a novelty that has not been done by other research, but needs to be improved in methods and discussion.

Reviewer 2 Report
This study investigated the associations among health status, occupation, and occupational injuries or diseases. It is original because hierarchical generalized linear model (HGLM) and occupational subcategories were used as the authors mentioned. However, the manuscript needs modification to be improved.
1. First of all, the limitation of this study should be described. Although demographic factors, physical and mental health status, and 28 subcategories of occupation were analyzed with HGLM based on population data, the whole medical and mental disorders were not included in this study and the analysis with subcategories of occupation is still ambiguous to verify the association between specific occupation and occupational injuries or diseases.
2. In Figure 1, or in an additional Table or Figure, it seems better to present the whole specific variables of occupational categories and subcategories, and physical and mental health, for readers to understand at a glance.
3. In line 69, specific ‘benefit codes for occupational injuries and disease’ should be described.
4. Insured amount seems to be proportional to income level. Then, describe the meaning of insured amount in Materials and Methods.
5. In Table 1, whether each variable was significant or not was presented. But, it seems better to present whether each variable was a significantly higher or lower risk, instead of check marks.
6. In line 128, ‘NT$57,801 or higher had 0.264-fold higher risks’, 0.264-fold higher is a lower risk less than 1. Check this. And if NT$57,801 or higher had a higher risk in line 128, in line 213, ‘those with low income levels had a higher risk of occupational injuries and diseases’ should be modified.
Round 2
Reviewer 2 Report
The manuscript has been improved after authors’ revision. However, several comments have not been responded sufficiently.
1. It had better to mention the limitation of this study in Discussion (for example, the whole medical and mental disorders could not be included in this study, etc.). Reconsider the description or rebut this.
2. In line 137 (‘had 0.264-fold higher risks’), ‘0.264-fold’ is a ‘lower risk’ because ‘0.264’ is less than ‘1’. The value ‘0.264’ might be incorrect or ‘higher’ should be changed to ‘lower’. Or it might be ‘relative higher risk’ to ‘0.177-fold lower risk of NT$36,301-45,800’. The sentence in lines 135-137 is difficult to understand.
3. In Table 2, is ‘Pr>F’ p-value? Modify this.
Author Response
Responses to the comments of Reviewer #2
- It had better to mention the limitation of this study in Discussion (for example, the whole medical and mental disorders could not be included in this study, etc.). Reconsider the description or rebut this.
Response: In lines 209-210 (shown yellow highlights and red words)
- In line 137 (‘had 0.264-fold higher risks’), ‘0.264-fold’ is a ‘lower risk’ because ‘0.264’ is less than ‘1’. The value ‘0.264’ might be incorrect or ‘higher’ should be changed to ‘lower’. Or it might be ‘relative higher risk’ to ‘0.177-fold lower risk of NT$36,301-45,800’. The sentence in lines 135-137 is difficult to understand.
Response: In lines 137-138 (shown yellow highlights and red words)
3. In Table 2, is ‘Pr>F’ p-value? Modify this.
Response: In line 104 (shown yellow highlights and red words)